# First Line Systemic Treatment for MALT Lymphoma—Do We Still Need Chemotherapy? Real World Data from the Medical University Vienna

**DOI:** 10.3390/cancers12123533

**Published:** 2020-11-26

**Authors:** Barbara Kiesewetter, Ingrid Simonitsch-Klupp, Marius E. Mayerhoefer, Werner Dolak, Julius Lukas, Markus Raderer

**Affiliations:** 1Department of Medicine I, Division of Oncology, Medical University of Vienna, A-1090 Vienna, Austria; markus.raderer@meduniwien.ac.at; 2Department of Pathology, Medical University of Vienna, A-1090 Vienna, Austria; ingrid.simonitsch-klupp@meduniwien.ac.at; 3Department of Radiology, Memorial Sloan Kettering Cancer Center, New York, NY 10065, USA; mayerhom@mskcc.org; 4Department of Biomedical Imaging and Image-guided Therapy, Division of Nuclear Medicine, Medical University of Vienna, A-1090 Vienna, Austria; 5Department of Medicine III, Division of Gastroenterology and Hepatology, Medical University of Vienna, A-1090 Vienna, Austria; werner.dolak@meduniwien.ac.at; 6Department of Ophthalmology, Medical University of Vienna, A-1090 Vienna, Austria; Julius.lukas@meduniwien.ac.at

**Keywords:** extranodal lymphoma, MALT lymphoma, immunomodulatory treatment, chemotherapy

## Abstract

**Simple Summary:**

The Division of Oncology at the Medical University Vienna is a tertiary referral center for extranodal lymphoma of mucosa-associated lymphoid tissue (MALT)-type and currently oversees more than 400 patients with this diagnosis, including a relevant percentage of patients being treated with systemic therapies. In the current analysis, we present response and long-term data of 159 patients (not eligible for *Helicobacter pylori* eradication) treated upfront with systemic treatment focusing on chemotherapy- versus immunotherapy-based strategies. We show that despite higher response and complete remission rates for chemo- versus immunotherapy, there appears to be no difference in progression-free survival, thus suggesting comparable long-term results. Considering the biological background of MALT lymphoma—and particularly its high dependence on the tumor microenvironment—but also the favorable toxicity profile of assessed immunomodulatory agents such as IMiD lenalidomide or macrolide clarithromycin, we suggest that these data support further investigation of chemotherapy-free treatment concepts for MALT lymphoma.

**Abstract:**

There is no clear therapeutic algorithm for mucosa-associated lymphoid tissue (MALT) lymphoma beyond Helicobacter pylori eradication and while chemotherapy-based regimens are standard for MALT lymphoma patients in need of systemic treatment, it appears of interest to also investigate chemotherapy-free strategies. We have retrospectively assessed MALT lymphoma patients undergoing upfront systemic treatment, classified either as chemotherapy (=classical cytostatic agents +/− rituximab) or immunotherapy (=immunomodulatory agents or single anti-CD20 antibodies) at the Medical University Vienna 1999–2019. The primary endpoint was progression-free survival (PFS). In total, 159 patients were identified with a median follow-up of 67 months. The majority of patients had extragastric disease (80%), but we also identified 32 patients (20%) with Helicobacter pylori negative or disseminated gastric lymphoma. Regarding the type of first line treatment and outcome, 46% (74/159) received a chemotherapy-based regimen and 54% (85/159) immunotherapy including IMiDs lenalidomide/thalidomide (37%), anti-CD20-anitbodies rituximab/ofatumumab (27%), macrolides clarithromycin/azithromycin (27%) and proteasome inhibitor bortezomib (9%). Median PFS was 76 months (95%CI 50–102), and while the overall response (90% vs. 68%, *p <* 0.01) and the complete remission rate (75% vs. 43%, *p <* 0.01) was significantly higher for chemotherapy, there was no difference in PFS between chemotherapy (median 81 months, 95%CI 47–116) and immunotherapy (76 months, 95%CI 50–103, *p* = 0.57), suggesting comparable long-term outcomes. To conclude, our data show higher response rates with chemo- compared to immunotherapy, but this did not translate into a superior PFS. Given the biological background of MALT lymphoma, and the favorable toxicity profile of novel immunomodulatory treatments, this should be further investigated.

## 1. Introduction

Extranodal marginal zone B-cell lymphoma of the mucosa-associated lymphoid tissue (MALT lymphoma) is a distinct type of B-cell lymphoma characterized by a heterogenous infiltrate consisting of small B-cells in epithelial structures and the marginal zone of secondary lymphoid follicles and a typical, relatively “non-descript” immunophenotype of CD20+CD5-CD10-cyclinD1- [1]. The stomach represents the most common localization documented in up to 50% of cases, but the disease may develop in mucosa-associated lymphoid tissues across all organs frequently involving the ocular adnexa, the lung, the thyroid and the parotid glands [2]. The pathogenesis of MALT lymphoma is connected to chronic inflammation as exemplified by Helicobacter (H.) pylori associated gastritis and development of gastric MALT lymphoma, but also autoimmune disorders may trigger MALT lymphoma with particularly Sjögren’s syndrome and Hashimoto’s disease being relevant comorbidities in these patients. This distinct pathomechanism is based on a multistep process of persistent autoantigenic stimulation of marginal zone B-cells co-orchestrated by (*H. pylori*) specific T-cells, evasion of further immune effector cells and an increase in proinflammatory cytokines [3,4]. A central molecular pathway is nuclear factor (NF)-kappa B, which interestingly, is not only activated by the above-mentioned factors of inflammatory processes, but also through MALT lymphoma specific genetic aberrations, e.g., t(11;18)(q21;21) BIRC3/MALT1, suggesting an advanced interaction of lymphoma cells with their microenvironment as well as immunologic and oncogenic factors [5].

Targeting *H. pylori* with antibiotic eradication treatment is one of the most successful examples of personalized medicine. By eliminating the bacteria in *H. pylori* positive gastric MALT lymphoma, long-term remissions in 60–80% of patients have been reported and eradication treatment constitutes the clear standard of care in this situation [2,6,7]. The treatment algorithm is less clear in extragastric and disseminated disease and given the generally indolent clinical course of MALT lymphoma with 5-year survival rates >90%, the individual risk–benefit ratio is an important factor in choosing the right therapy. While radiotherapy constitutes an effective and approved treatment for localized extragastric or *H. pylori* refractory gastric disease, chemotherapy based-treatment in line with other indolent B-cell lymphomas is the preferred approach for patients in need of systemic treatment [6,8,9]. Commonly applied regimens include chlorambucil +/− rituximab (R) and R-bendamustine, a combination initially implemented for the treatment of follicular lymphoma [10,11]. Both regimens result in high overall response rates (ORR) and favorable long-term outcomes.

Despite these positive data on R-chemotherapy, there has been increasing interest in recent years to evaluate chemotherapy-free immunomodulatory treatment strategies for MALT lymphoma [12]. This is based on the fact that MALT lymphoma is an indolent disease requiring the least toxic individual treatment, and also on the observation that MALT lymphoma is highly dependent on its microenvironment, thus predisposing this disease for immunomodulatory concepts. While R-monotherapy has already been implemented for MALT lymphoma, more recent investigations have assessed the proteasome inhibitor bortezomib, Bruton’s kinase inhibitor ibrutinib and actual immunomodulatory compounds such as the macrolide antibiotic clarithromycin and IMiD-based treatment with lenalidomide +/− R [8,13,14,15,16,17,18,19,20]. Most available data, however, derive from phase II pilot trials and in contrast to well established classical cytostatic treatment strategies, there is a lack of long-term “real world” data.

The Division of Oncology at the Medical University of Vienna is a tertiary referral center for extranodal lymphoma of MALT-type and has a database of currently 400+ patients registered with this diagnosis, including a relevant percentage of patients being treated systemically including immunomodulatory therapies. In the current analysis, we present response and long-term data of patients treated upfront with systemic therapy focusing on chemotherapy-based versus immunotherapeutic concepts.

## 2. Methods

We have retrospectively assessed all patients diagnosed, staged and treated for MALT lymphoma at the Medical University of Vienna, Department of Medicine I, Division of Oncology, between 1999 and 2019. Only patients treated upfront with systemic treatment classified either as chemotherapy-based (i.e., classical cytostatic agents +/− R) or immunotherapy (i.e., CD20-targeting strategies or immunomodulatory compounds) were included in this analysis. All histological diagnoses were (re-) established according to the most recent version of the WHO classification of hematopoetic and lymphoid tissues including adequate immunophenotyping as outlined for MALT lymphoma, demonstration of light chain restriction and assessment of plasmacytic differentiation [1]. Patients and clinical data identified were extracted from electronic and/or paper-based records routinely stored at the department. The current analysis had been approved by the local ethical board of the Medical University of Vienna (EK-No.: 791/2011, September 2019).

### 2.1. Patients Characteristics

Data extracted were basic patient characteristics including date of diagnosis, age at initial diagnosis, sex and performance status; disease specific characteristics including stage of disease according to Ann Arbor and/or the Lugano classification in case of gastric MALT lymphoma and localization of primary disease; histological features including presence or absence of plasmacytic differentiation and disease-specific laboratory values such as lactate dehydrogenase levels (LDH), beta-2-microglobulin and paraproteinemia. All patients were routinely screened for MALT lymphoma-relevant comorbidities, i.e., *H. pylori* infection, hepatitis B/C virus and autoimmune disorders, and subsequent results were documented for this investigation. In addition, the MALT-IPI risk score was retrospectively assessed in all patients with complete data available (risk factors: LDH, age 70-plus, Ann Arbor stage IIIE/IV) and classified for low risk (no risk factor), intermediate risk (one risk factor) and high risk (≥2 risk factors) [21].

### 2.2. Treatment, Response Assessment and Follow-Up

Exact type of treatment, start/stop date, date of best response and date of progression were documented. Staging at initial diagnosis and prior to treatment was performed according to current guidelines including at least one systemic imaging at primary diagnosis and—based on these findings—organ-dependent imaging for further follow-up. During active treatment, response assessment was routinely performed every three months, followed by increased intervals of 6–12 months after completion of therapy with ongoing clinical check-ups at our department, allowing an unbiased longitudinal follow-up. Response assessment was performed by radiological criteria in case of extragastric disease only, i.e., complete remission (CR), partial remission (PR), stable disease (SD) and progressive disease (PD) and supplemented by histological GELA response criteria in gastric MALT lymphoma [22]. Progression-free survival (PFS), time-to-progression (TTP) and time-to-next-treatment (TTNT) were calculated as defined in the revised response criteria published by Cheson et al. [23]. Finally, date of last visit for assessment of follow-up times, overall survival (OS) and occurrence of transformation to aggressive lymphoma were documented.

### 2.3. Stastical Analysis

Data Statistical computations were performed using IBM Statistics for Mac OS version 26.0 (IBM, Armonk, NY, USA). Metric data were described displaying median, range (minimum/maximum) and interquartile range (IQR). Percentages and absolute frequencies were presented for categorical variables. Chi-square and Fisher’s exact test were used for analyzing associations of binary variables. Estimated PFS and OS curves were plotted by use of the Kaplan–Meier method and medians including confidence intervals (CI) presented, in addition log-rank test was applied to compare two groups. A proportional cox-regression model was used to assess impact of MALT-IPI factors on PFS and OS. *p*-values (two-sided) < 0.05 were accepted as statistically significant.

## 3. Results

A total of 412 patients were treated for MALT lymphoma at our department 1999 to 2019, and we could identify 159 patients (38.5%) who received first line systemic treatment according to the criteria defined in the methods section. These patients were included for further analysis. In this group, the median follow-up time was 66.5 months (IQR 33.2–100.6). There was a slight surplus of female (90/159, 57%) versus male patients (69/159, 43%), resulting in a female-to-male ratio of 1.3. The median age at initial diagnosis was 65 years (range 25–88). In terms of primary localization, ocular adnexal MALT lymphoma constituted the most frequent primary lymphoma manifestation accounting for 33%, followed by gastric MALT lymphoma in 20%, lung in 17%, parotid in 11%, skin/soft tissues in 4%, breast in 4%, colon in 3% and (non-parotid) salivary glands in 3%. In addition, rare localizations such as the adrenal gland (*n =* 2), the liver (*n =* 2), the thyroid (*n =* 2), the tonsils (*n =* 1), the bladder (*n =* 1) and the kidney (*n =* 1) were documented.

Regarding stage of disease, 43% presented with localized disease (=Ann Arbor IE), 21% had local lymph node involvement (Ann Arbor IIE), 5% distant lymph node involvement (Ann Arbor IIIE) and 31% presented with upfront multi-organ involvement (Ann Arbor IV). In 94% of patients (149/159), all data for calculation of MALT-IPI scores were available and stratified; 39% into the low risk group, 46% into intermediate risk and 15% had more than one risk factor, i.e., high risk according to published criteria [21]. In line with current guidelines, all 32 gastric MALT lymphoma patients treated upfront with systemic therapy had been non-eligible for *H. pylori* antibiotic eradication, and were thus either *H. pylori* negative (23/32, 72%) or had symptomatic disseminated disease with need for systemic treatment (9/32, 28%). See Table 1 for more detailed baseline characteristics.

### 3.1. Treatment Characteristics

First line treatment of 159 patients treated with systemic therapy was chemotherapy-based in 46% (74/159) and classified as immunotherapy in 54% (85/159). Only patients receiving exclusively systemic treatment have been included; none of the patients had received prior or concomitant radiotherapy. In detail, the most commonly applied chemotherapeutic regimens included purine analogue-based therapy +/−R (23/74, 31%), CHOP (i.e., cyclophosphamide, doxorubicin, vincristine and prednisone) +/−R (21/74, 28%), R-bendamustine (18%, 13/74) and chlorambucil-based therapy (11%, 8/74). All patients treated with purine analogs (cladribine, fludarabine) were part of clinical trials previously published but were followed for a more prolonged time than in the initial publications. In the immunotherapy group, IMiDs thalidomide and lenalidomide +/− R constituted the largest group of patients (37%, 31/85), followed by monotherapy with CD20-antibodies rituximab or ofatumumab (27%, 23/85), macrolide antibiotics clarithromycin or azithromycin (27%, 23/85) and the proteasome inhibitor bortezomib (9%, 8/85). Again, these patients had been included in clinical trials for thalidomide, lenalidomide, ofatumumab, azithromycin and bortezomib. Given the fact that chemotherapy-free treatment is a more recent focus of our center, 69% of the chemotherapy-group received treatment before 2009 versus 31% thereafter, reaching a significant difference (*p <* 0.001). This is underlined by the median follow-up being longer for the chemo- versus the immunotherapy group (87.3 months versus 57.4 months, *p =* 0.002). Additionally, R was part of the chemotherapy regimen in all but two patients treated in the last decade, while chemotherapy without R was applied more frequently for patients treated earlier, again highlighting the development of the treatment landscape over time. In terms of clinical characteristics, there were no differences between treatment groups, despite a trend for increased LDH-levels in the chemo-group (documented in 11% versus 5% for immunotherapy, *p =* 0.18) and significantly more patients with disseminated disease (45% versus 28%, *p =* 0.032) in the chemo-collective if compared to patients receiving immunotherapy. The number of MALTI-IPI intermediate/high risk patients was equal (62% versus 60%, *p =* 0.815). For more detailed information regarding treatment characteristics see Table 2.

### 3.2. Objective Response to Treatment

Data on objective best response to first line systemic treatment were available in 96% of patients (153/159), and response was classified as CR in 58%, PR in 21%, stabilization of disease in 18% and primary progressive disease in only 3%. Per cohort, the best response in the chemotherapy group was CR in 75%, PR in 16% and SD in 3%, while 7% had PD; for immunotherapy CR in 43%, PR in 43% and SD in 32% with no patients experiencing primary progression. In direct comparison, the overall response rate (ORR) (90% versus 68%, *p =* 0.001) and the rate of CR (75% versus 43%, *p <* 0.001) was significantly higher for chemotherapy than immunotherapy. However, none of the patients with immunotherapy showed disease progression during active treatment versus 7% in the chemo-group.

Time to best response did not differ statistically (median 4.9 months for chemo and 5.3 months for immunotherapy, *p =* 0.216), but the absolute range for time to the best response was longer for immunotherapy (up to 66 months until best response) versus chemo (up to 21 months until best response).

### 3.3. Progression-Free Survival

The median estimated PFS for the entire collective was 76.4 months (95%CI 50.4–102.4); median PFS was 81.2 months (95%CI 46.6–115.8) for chemotherapy-based treatment and 76.4 months (95%CI 49.6–103.2) for immunotherapy with no significant difference as compared by log-rank test (*p =* 0.566). This difference remained statistically non-significant following multivariate correction for MALT-IPI factors, with elevated LDH identified as the only independent predictor of worse PFS (*p =* 0.017). See Figure 1 for the respective curve. There was also no impact of the primary treatment for the subgroup of gastric patients only (*p =* 0.502) and extragastric only (*p =* 0.457), respectively (see Figure 2). Kaplan–Meier curves of further frequent primary localizations including MALT lymphoma of the ocular adnexa, lung and parotid glands are provided in Appendix A. There was no difference for localized (stage I/II, *p =* 0.896) and disseminated (stage III/IV, *p =* 0.472) disease in terms of PFS comparing chemo- and immunotherapy (see Figure 2). Finally, we did not observe a difference within the specific MALT-IPI groups between chemotherapy versus immunotherapy, i.e., the low-risk patients (*p =* 0.597), the intermediate risk group (*p =* 0.585) and the high-risk group (*p =* 0.38). A complete table of univariate analysis for PFS and OS, according to clinical features, is shown as Appendix A.

The effective relapse rate was 43% for the entire collective with a non-significant trend for more frequent relapses documented in the chemo group (39% versus 50%, *p =* 0.15). In line with this finding, there were numerically more patients in need of second line treatment in the chemo group (47% versus 34%, *p =* 0.09). Median time to progression (TTP) in patients experiencing a relapse was 22.2 months (IQR 13.3–50.4) and TTP did not differ between patients previously treated with chemo- versus immunotherapy (18.7 versus 24.8 months, *p =* 0.85). Median time to next treatment (TTNT) for patients in need of second line therapy was 18.7 months and was again not statistically different between patients previously treated with chemo- versus immunotherapy (18.7 versus 14.6, *p =* 0.313). TTP and TTNT Kaplan–Meier plots are shown in Figure 3.

### 3.4. Long-Term Outcome

As expected, long-term survival was excellent with a 5-year survival rate of 88% in the overall collective. Interestingly, the overall survival was significantly better in the immunotherapy group versus the chemo-group (5-year OS 91% versus 85%, *p =* 0.03), and this remained statistically significant after correction for MALT-IPI factors with age 70-plus identified as a single independent factor for worse OS (*p =* 0.038). However, this finding clearly remains exploratory and is potentially biased due to the significantly longer follow-up time in the chemotherapy group. The rate of transformation to aggressive B-cell lymphoma was low at 3% (5/159) and was numerically, but not relevantly, higher in the chemotherapy group (4% versus 2%). Four of five patients with transformed disease, however, died due to lymphoma. Finally, the number of (non-transformation) secondary malignancies also does not appear to be elevated over the general population with 9% (15/159) reported in total. In relative numbers, secondary malignancies were documented in 5% (*n =* 4) for immunotherapy and 15% (*n =* 11) for chemotherapy, but this again needs to be interpreted with caution in view of the longer follow-up in the chemo-group. Kaplan–Meier curves for OS and lymphoma-specific survival are shown in Figure 4.

## 4. Discussion

MALT lymphoma is characterized by an indolent clinical course but up to one third of patients present with disseminated disease and relapses including distant recurrence are observed in a significant amount of patients, even after successful local treatment such as radiotherapy [24,25,26]. This underlines the potentially systemic nature of MALT lymphoma and also the feasibility to evaluate systemic treatment approaches in localized disease, and in fact various—albeit retrospective—analyses have suggested systemic therapy as a valid alternative also in stage I/IIE. In the current analysis, we present data on 159 patients, treated upfront with systemic therapy with a focus on chemotherapy-based versus immunotherapeutic treatment. While the caveat of this analysis is the diversity of treatment strategies and the fact that a number of patients have been treated within clinical trials, the strengths are the single center approach and the uniform staging and follow-up protocol, assuring consistent and unbiased longitudinal data [27].

Chemotherapy-based treatment still constitutes the current standard of care based on available long-term data in most indolent lymphomas, although the use of rituximab plus lenalidomide has been shown as a potential alternative to chemotherapy in follicular lymphoma [28]. Numerous systemic treatment strategies, however, have been investigated for the treatment of—mostly symptomatic—MALT lymphoma, without a clearly defined standard so far [8]. The use of R-chlorambucil is supported by results of the IELSG-19 trial, a three arm-randomized study that included a total of 454 patients treated with R-chlorambucil or the respective monotherapies [10]. The primary endpoint was event-free survival (EFS) of chlorambucil +/− R and the published long-term data seemed to confirm the superior efficacy of the combination, at least in terms of 5-year EFS of 68% versus 51% (HR 0.54, 95%CI 0.37–0.77). The initial response rates were high at 95% versus 86%, and 78% for the R-mono arm. However, given the equally excellent OS of 90% at 5 years in all three treatment arms, this study also provides evidence for use of chlorambucil and R-monotherapy if combination therapy is not feasible. R-bendamustine was evaluated in phase III studies for follicular lymphoma including also subgroups of marginal zone lymphoma patients and has been considered standard treatment in these patients [29,30]. For MALT lymphoma, frontline data from a Spanish phase II study on 60 patients treated with a response-based schedule justify the use of this combination, especially as the ORR of 100% and long-term results are encouraging [11]. Further chemotherapeutic regimens previously tested and also included in the current analysis are purine analogs and CHOP-based therapy, which have subsequently been replaced by these less toxic treatments in spite of impressive activity [8].

The concept of immunomodulatory treatment for MALT lymphoma, however, appears obvious in view of the complex interplay between antigenic stimulation, the (ineffective) immune response and the microenvironment in the pathogenesis of the disease [12]. Substances evaluated in this context include IMiDs and macrolide antibiotics, but also compounds that are more immuno-therapeutic than -modulatory, such as CD20-antibodies and bortezomib. The IMiD lenalidomide was tested in pilot trials both as monotherapy and in combination with R, but the latter was more effective resulting in an ORR of 80% and >50% CRs in 46 patients [17,18]. In addition, recently reported long-term results suggest a median PFS of 60–70 months for marginal zone lymphoma of various sites treated with R-lenalidomide with low rates of toxicity [31,32]. Consequently, the European Society for Medical Oncology (ESMO) 2020 guidelines have included R-lenalidomide as a potential treatment strategy for relapsed and refractory MALT lymphoma [6]. Furthermore, macrolide antibiotics and clarithromycin in particular, appear promising given their combined pleiotropic immunomodulatory prosperities including activation of T-cells but also antimicrobial effects [33]. The distinct antitumor efficacy of the macrolide clarithromycin has been established in two phase II trials and one large retrospective series, including only patients in whom infectious triggers such as *H. pylori* and *Chlamydophila psittaci* had previously been ruled out [14,15,16]. Reported response rates were in the range of up to 50% with a 3-year PFS of 50%, even in heavily pretreated patients. Bortezomib-based therapy has also given high response rates, but the high neurotoxicity of the intravenous applications has not led to further trials with this agent [19,20]. Finally, anti-CD20 antibody monotherapy constitutes a well-tolerated approach, and while second generation compounds such as ofatumumab rely only on pilot data, R-monotherapy is an accepted treatment for low-tumor burden disease or patients with comorbidities, not least due to the phase III data from the IELSG trial (ORR for R-monotherapy arm 78%, CR 56%, 5-year EFS 50%) [10,34].

The results presented in the current analysis confirm previous data showing that objective response rates are higher with chemotherapy-based treatment if compared to immunotherapy, and we report a statistically significant difference for the ORR (90% versus 68%, *p =* 0.001) and CR-rate (75% versus 43%, *p <* 0.001), which might be clinically relevant in the presence of symptoms related to lymphoma localization. However, patients with a high tumor burden constitute a minority in this indolent disease and extrapolating from data in gastric MALT lymphoma with residual disease, there appears no clear indication for treating persistent disease in order to force a CR in the absence of documented progression or clinically relevant symptoms following treatment [6,7,35]. Thus, CR rates might be of minor relevance and PFS or POD24 (=progression of disease at 24 months) are potentially better surrogate parameters for long-term outcome in this indolent disease with a lack of mature OS results for most available trials [36]. As previously reported, median PFS for the entire cohort was excellent at >5 years and did not differ between chemotherapy (median 81 months) and immunotherapy (76 months). These data support the hypothesis that long-term outcomes following immunotherapy may be considered equal to chemotherapy-based treatment due to achieving a balance between lymphoma and the immune system allowing long term control of the disease. This is further underlined by the OS data found in our analysis, with all the caveats of a retrospective series.

The high rate of patients with localized disease treated with systemic treatment in this analysis (43% stage I and 21% stage II) deserves special consideration, as radiotherapy has been reported as highly active therapy, especially in terms of local control. However, given the previously discussed potentially systemic nature of this disease and the fact that radiotherapy in conventional doses is associated with side effects particularly in radiation-sensitive tissues like the ocular adnexa (cataract, red eye syndrome) or the gastrointestinal mucosa (strictures, gastroparesis), we suggest that systemic treatment is feasible and also active for patients with localized MALT lymphoma [26,37]. This is supported by our finding that PFS for each treatment did not differ in localized and disseminated disease subgroups (*p*-values non-significant, Figure 2). Furthermore, we did not find a difference regarding PFS and treatment in view of primary localization (gastric/extragastric, *p*-values non-significant). Interestingly, ongoing discussions on the optimal dosage of radiotherapy such as reports on the high activity of “ultra-low” doses of 2 × 2 Gy might result in a potential paradigm shift in this regard in the near future [38].

However, one weakness of this analysis is that we do not present a detailed comparison of toxicities, thus, while recently published results of pilot trials and also first long-term data suggest a favorable toxicity profile for immunotherapy with low hematologic and non-hematologic toxicity for lenalidomide-based treatment and de facto absent toxicity for clarithromycin, this remains speculative within the here presented data.

## 5. Conclusions

The current series shows that the higher response rates induced by chemotherapy did not translate into longer PFS in our patient cohort. Thus, in view of pathogenetic considerations and the low toxicity profile of most immunotherapeutic compounds, development of chemotherapy-free treatment strategies should be further encouraged for MALT lymphoma. Established regimens like R-lenalidomide for relapsed disease or R-monotherapy or clarithromycin in cases of low-tumor burden constitute a feasible and safe approach outside of clinical trials. The value of systemic treatment for localized and asymptomatic disease should be further investigated in prospective trials.

## Figures and Tables

**Figure 1 cancers-12-03533-f001:**
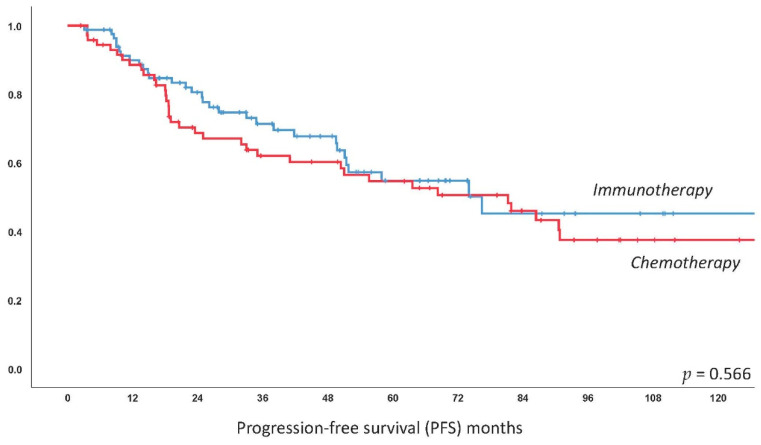
Kaplan–Meier curve for estimated progression-free survival in MALT lymphoma patients treated with immunotherapy versus chemotherapy. *X*-axis follow-up in months, *Y*-axis cumulative progression-free survival.

**Figure 2 cancers-12-03533-f002:**
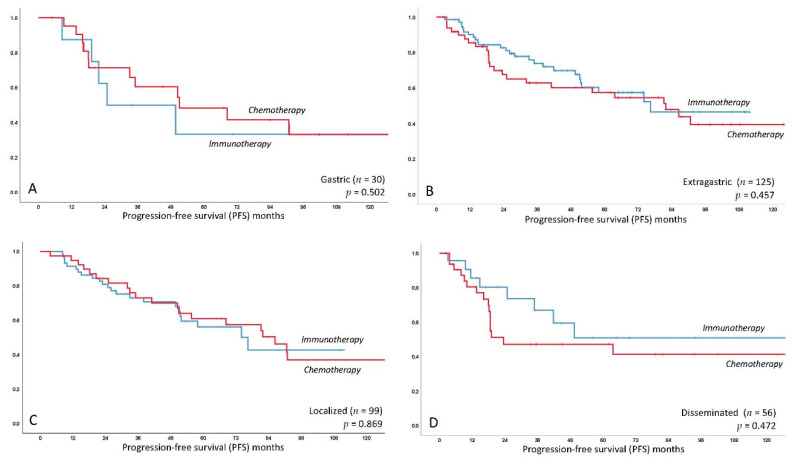
Kaplan–Meier curve for estimated progression-free survival in MALT lymphoma patients treated with immunotherapy versus chemotherapy for the following subgroups: (**A**) gastric MALT lymphoma patients; (**B**) extragastric MALT lymphoma patients; (**C**) localized MALT lymphoma patients; (**D**) disseminated MALT lymphoma patients. *X*-axis follow-up in months, *Y*-axis cumulative progression-free survival.

**Figure 3 cancers-12-03533-f003:**
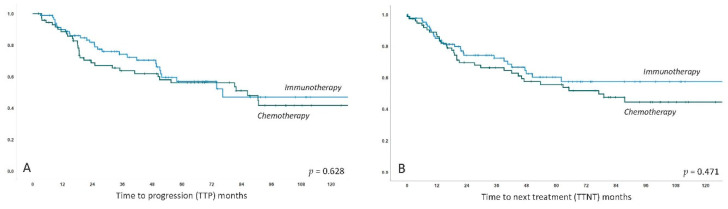
Kaplan–Meier curves for estimated time to progression (TTP) (**A**) and time to next treatment (TTNT) (**B**) in MALT lymphoma patients treated with immunotherapy versus chemotherapy. *X*-axis follow-up in months, *Y*-axis cumulative probability of events.

**Figure 4 cancers-12-03533-f004:**
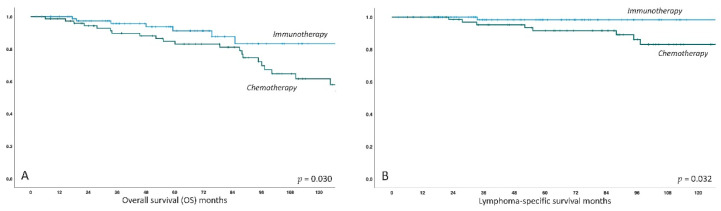
Kaplan–Meier curves for estimated overall survival (**A**) and lymphoma-specific survival (**B**) in MALT lymphoma patients treated with immunotherapy versus chemotherapy. *X*-axis follow-up in months, *Y*-axis cumulative survival.

**Table 1 cancers-12-03533-t001:** Baseline features of 159 patients treated with systemic therapy and comparison of the immunotherapy versus the chemotherapy cohort with respect to patient’s characteristics.

Feature	Entire Collective	Immunotherapy	Chemotherapy	*p*-Value
	*n* = 159	*n* = 85	*n* = 74	
Sex				
Female	57% (90/159)	61% (52/85)	51% (38/74)	*p* = 0.212
Male	43% (69/159)	39% (33/85)	49% (36/74)	
Age (median)	65 years	66 years	64 years	*p* = 0.764
Age 70+	33% (53/159)	37% (31/85)	30% (22/74)	*p* = 0.368
Primary localization				
Gastric	20% (32/159)	11% (9/85)	31% (23/74)	*p* = 0.001
Extragastric	80% (127/159)	89% (76/85)	69% (51/74)	
Stage of disease				
Localized (I/IIE)	64% (102/159)	72% (61/85)	55% (41/74)	*p* = 0.032
Disseminated (III/IVE)	36% (57/159)	28% (24/85)	45% (33/74)	
MALT-IPI				
Low risk	39% (58/149)	40% (33/83)	38% (25/66)	*p* = 0.815
Intermediate/high risk	61% (91/149)	60% (50/83)	62% (41/66)	
Further clinical features				
Autoimmune disorder *	36% (50/141)	32% (27/84)	40% (23/57)	*p* = 0.371
Plasmacytic diff.	38% (47/125)	35% (26/75)	42% (21/50)	*p =* 0.407
LDH > ULN	7% (11/149)	5% (4/83)	11% (7/66)	*p =* 0.180
Performance status > 1	5% (8/156)	4% (3/85)	7% (5/71)	*p =* 0.472
Viral hepatitis (B/C)	2% (3/132)	0% (0/79)	6% (3/53)	*p =* 0.063
Beta2-micorglobulin > ULN	45% (59/131)	38% (29/77)	56% (30/54)	*p =* 0.043
Paraproteinemia	34% (36/107)	29% (19/65)	40% (17/42)	*p =* 0.229
Time to treatment (median)	2.0 months	2.2 months	1.8 months	*p =* 0.013
Follow-up time (median)	66.5 months	57.4 months	87.3 months	*p =* 0.002

* includes Sjogren’s syndrome (*n =* 21), Hashimoto’s thyroiditis (*n =* 9), psoriasis arthritis (*n =* 1), systemic lupus erythematous (*n =* 2), leukocytoclastic vasculitis (*n =* 1), unspecific elevation of auto-antibodies and/or rheumatoid factors (*n =* 16). Abbreviations in chronological order: MALT = mucosa-associated lymphoid tissue; IPI = international prognostic index; LDH = lactate dehydrogenase levels; diff. = differentiation; ULN = upper limit of normal.

**Table 2 cancers-12-03533-t002:** Treatment characteristics of 159 MALT lymphoma patients treated with systemic therapy.

Treatment Characteristics	
**First line immunotherapy**	**54% (85/159):**
IMiDs +/−R	37% (31/85) (+R in 18/31)
CD20-antibody monotherapy	27% (23/85)
Macrolides	27% (23/85)
Bortezomib	9% (8/85)
**First line chemotherapy +/− R**	**46% (74/159) (+ R in 42/74)**
Purine analogs +/−R	31% (23/74) (+ R in 15/23)
CHOP-based +/−R	28% (21/74) (+ R in 13/21)
Bendamustine + R	18% (13/74) (+ R in 13/13)
Chlorambucil +/−R	11% (8/74) (+ R in 1/8)
Other	12% (9/74)
**Per localization (primary sites >10%)**	
Ocular adnexal MALT lymphoma (*n =* 52, 33%)	
First line immunotherapy	29% (15/52)
First line chemotherapy	71% (37/52)
Gastric MALT lymphoma (*n =* 32, 20%)	
First line immunotherapy	72% (23/32)
First line chemotherapy	28% (9/32)
Pulmonary MALT lymphoma (*n =* 27, 17%)	
First line immunotherapy	44% (12/27)
First line chemotherapy	56% (15/27)
Parotid gland MALT lymphoma (*n =* 18, 11%)	
First line immunotherapy	44% (8/18)
First line chemotherapy	56% (10/18)

Abbreviations in chronological order: R = rituximab; CHOP = cyclophosphamide, doxorubicin, vincristine and prednisone.

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
