# Peer review of "First Line Systemic Treatment for MALT Lymphoma—Do We Still Need Chemotherapy? Real World Data from the Medical University Vienna"

_cancers, 2020, doi:10.3390/cancers12123533_

Round 1
Reviewer 1 Report
This is a timely and interesting paper describing the comparison between chemotherapy- and immunotherapy-based regimens against MALT-lymphoma. I agree with the authors that the PFS or long-term outcome is more important than short-term response in the case of MALT lymphoma treatment. Although most data presented seem to support the authors’ conclusions, the following point should be addressed.
Radiotherapy is commonly used in combination with either chemotherapy or immunotherapy. The authors should describe whether some of the 159 patients received radiotherapy or not. If some of them received radiotherapy, its effect on the outcome should be evaluated.
Reviewer 2 Report
The study performed by Barbara Kiesewetter et al. analyzed the response and long-term data of 159 patients with MALT lymphoma, and demonstrated comparable long-term results between chemo-based therapy and immunotherapy, especially for the progression-free survival, despite higher response and complete remission rates for chemotherapy. Indeed, chemo-free therapy is the current trend for many types of hematological malignancies, and the data presented herein is of value in developing the chemo-free standards for MALT lymphoma. Here are some comments:
- Since the aim of the current study is to compare outcome of chemotherapy and immunotherapy in MALT lymphoma treatment, the issue of toxicity is inevitable. The study should also compare the toxicity/side effect between the two treatments.
- Has any long-term toxicity/side effect, such as early menopause and secondary tumor, been identified in either group? Did the chemotherapy result in more frequent long-term toxicity or side effect than the immunotherapy? Chemotherapy associated long-term toxicity/side effect is particularly concerned in young patient. Did the young patients show comparable outcome to the two types of treatment?
- Did patients with different MALT-IPI group all show comparable outcome to the two types of treatment? As the overall response and the complete remission rate is higher in chemotherapy group, whether chemotherapy is still favorable for the high-risk group?
- Despite the comparable PFS, did the two groups share the same risk factors?
- A typo was identified: There was a slight surplus of female (90/159, 57%) versus male patients (69/159, 43%), resulting in a female-to-male ratio of 1:1.3.
Reviewer 3 Report
GENERAL COMMENTS
The authors analyze a relatively large series of 159 patients with extranodal marginal zone lymphoma (EMZL), who were in need of systemic first-line therapy, aiming to compare the outcomes of patients treated with (immuno)chemotherapy versus immunotherapy or immunmodulation in the context of chemo-free approaches. Many patients had participated in clinical trials. Although chemo-based approaches provided statistically better response rates, progression free Survival (PFS) and overall survival (OS) were not affected. Although the question is interesting, the study suffers from major methodological limitations.
SPECIFIC COMMENTS
Major Comments
- The major limitation is the extremely heterogenous treatment approaches, which were included in the two groups compared in this study. In the chemotherapy group, not only chemotherapy regimens were very different (purine analogues, CHOP, bendamustine, chlorambucil and others) but also the use of rituximab was variable. The percentage of patients that received rituximab with each regimen should be reported, but still treatment remains very heterogenous and rituximab-based chemoimmunotherapy (R-chlorambucil) provided better PFS than chlorambucil monotherapy or rituximab monotherapy in the IELSG-19 trial. The variable use of rituximab in the chemo-group results to a suboptimally treated population according to current standards in terms of PFS and may have affected the inability to demonstrate and PFS increase despite better response rates.
- The same issues are applicable for the immune (modulatory)therapy group as well. Treatment was highly diverse ranging from macrolides or bortezomib monotherapy to the well known rituximab (or ofatumumab) monotherapy or IMiDs with the latter given either as monotherapy or in combination to rituximab. All this variability renders any comparison of questionable value.
- Disease localizations were also very heterogenous, as expected for this disease. A more detailed comparison of treatment approaches according to the disease localization would be useful. Furthermore, patients’ characteristics according to treatment group and the relevant comparisons should be presented in an extended table, further to table 1 and the brief data reported in the text.
- Survival curves should be provided. The causes of death are also of particular interest because unrelated deaths may obscure differences between groups as well as the identification of prognostic factors. Unrelated deaths may also affect to a lesser extent PFS. TTP and TTNT curves should also be provided.
- The results of univariate analysis should be described in a table, including the PFS and probably OS or other endpoints by primary site. The same is applicable for multivariate analysis.
Minor Comments
- Since watch and wait strategy may be implemented for some patients with EMZL, the time-to-initial treatment should be provided according to treatment group as a baseline information.
- Please briefly report the associated autoimune disorders. Hepatitis status, b2-microglobulin and paraproteinemia are mentioned in “Methods” but not in the results.
Round 2
Reviewer 2 Report
The authors have replied to the comments properly.
Author Response
Thank you very much for this reply.
Reviewer 3 Report
GENERAL COMMENTS
The revised version of the manuscript has been considerably improved and most comments have been adequately addressed. Methodological limitations are inherent to this analysis; thus they inevitably remain as concerns. There are still some issues to be corrected.
SPECIFIC COMMENTS
Major Comments
- In table 1 the meaning of the p-value is unclear. I suppose that it represents the comparison between chemo and immunotherapy with respect to patients’ characteristics. However, the authors state that univariate analysis has been included in this table. This appears very unclear. The results of univariate analysis (PFS/OS) according to patients’ characteristics and major disease localizations are worthy to be included either in the main manuscript or as supplementary material. Median values would be more relevant than means for age, time to treatment and follow-up times in table 1. The numbers of patients in the columns “immunotherapy” and “chemotherapy” should be provided, as done for the entire series in the left column.
Minor Comments
In table 2, the numbers and percentages for gastric MALT lymphoma in the right column have been misplaced.
